# Alpha-Cyclodextrin Attenuates the Glycemic and Insulinemic Impact of White Bread in Healthy Male Volunteers

**DOI:** 10.3390/foods9010062

**Published:** 2020-01-07

**Authors:** Albert Bär, Ioannis Diamantis, Werner P. Venetz

**Affiliations:** 1Bioresco Ltd., 4054 Basel, Switzerland; 2Gastro Center, 2502 Biel, Switzerland; dr.diamantis@hin.ch; 3Datagen AG, 3930 Visp, Switzerland; werner.venetz@datagen.net

**Keywords:** alpha-cyclodextrin, dietary fiber, glycemic response, insulinemic response, amylase, digestion

## Abstract

The demonstration of a physiological benefit has recently become an indispensible element of the definition of dietary fibers. In the here-reported pilot study, the effect of alpha-cyclodextrin (alpha-CD) on the postprandial glycemic and insulinemic effect of starch was examined. Twelve fasted, healthy male volunteers received, on three subsequent days, a test breakfast consisting of (A) 100 g fresh white bread (providing 50 g starch) and 250 mL drinking water, (B) the same bread with a supplement of 10 g alpha-CD dissolved in the drinking water, and (C) 25 g alpha-CD dissolved in drinking water. Capillary and venous blood was sampled before the breakfast and in regular intervals for a three-hour period thereafter. Glucose was determined in capillary blood and insulin in the plasma of venous blood samples. Breakfast (A) led to a rapid rise in blood glucose and insulin. In breakfast (B), alpha-CD reduced the areas under the curve of blood glucose and insulin significantly by 59% and 57%, respectively, demonstrating that alpha-CD inhibits and thereby delays starch digestion. Treatment (C) was not associated with a rise of blood glucose. Hence, alpha-CD complies with the current definition of dietary fiber in every respect.

## 1. Introduction

According to a “Guidance for Industry” that the US Food and Drug Administration (FDA) published in June 2018, isolated or synthetic non-digestible carbohydrates that are added to formulated foods may be declared in nutritional labelling as dietary fiber only if their use has a demonstrated beneficial physiological effect [1]. Attenuation of post-prandial blood glucose and/or insulin levels is a generally recognized beneficial health effect. It may be measured by assessing blood glucose and insulin levels for up to at least two hours after consumption of a standardized meal. However, only studies in which an isolated or synthetic non-digestible carbohydrate is added to a food (rather than substitutes for a food component) provide evidence of an independent, i.e., active glycemia-lowering effect [2].

Alpha-cyclodextrin (CAS 10016-20-3) was shown to be a dietary fiber in compliance with these requirements in a randomized, double-blind crossover study in which ten healthy subjects consumed boiled white rice (containing 50 g digestible starch) combined with 0 (control), 5, or 10 g alpha-cyclodextrin (alpha-CD). Measurement of plasma glucose in regular intervals over a two-hour postprandial period showed that the area under the curve (AUC) was negatively related to the alpha-CD dose and that the difference to the untreated controls was significant (*p* ˂ 0.05) for the mid- and high-dose alpha-cyclodextrin treatment [3]. In the here-reported pilot study, the hypothesis was tested that alpha-cyclodextrin lowers the glycemic and insulinemic response to starch ingested with white bread that has a similar high glycemic index to boiled white rice [4] and that therefore is suitable for detecting glycemia and the insulinemia modulating effects of dietary fibers [5,6]

## 2. Materials and Methods

### 2.1. Recruitment and Ethics

Twelve healthy male volunteers were recruited from the Heraklion University campus. The subjects were 23–24 years old, had a body weight of 69–85 kg (mean 74.2 kg), and appeared to be healthy on medical pre-study screening. None of them consumed a special diet, had a history of gastrointestinal or metabolic disease, or was a smoker. The here-reported study was conducted in accordance with the Declaration of Helsinki. Its protocol and execution were approved and signed by the Ethics Committee of the University Hospital Medical School, Heraklion Greece, P.O. Box 1393, GR-71409 Heraklion, Greece.

After having been informed by the study director (I.D.) about the purpose, procedures, risks, and benefits of the planned study, all participants signed an informed consent document. Monetary compensation was accorded to each participant after partial or full completion of the study.

### 2.2. Preparation of Test Meals

#### 2.2.1. Bread Preparation

Bread was prepared fresh every day in the early morning by the kitchen of the University Hospital of Heraklion, Greece. For obtaining 1 kg bread, ready for consumption, 500 g white strong flour, 35 g salt, and 7 g yeast were mixed in a large bowl. Ten milliliters of olive oil and 300–400 mL water were added and mixed well. The mixture was kneaded for about 10 min. Once the dough was satin-smooth, it was placed in a lightly oiled bowl and covered with cling film. It was set to rise for 1 hour during, which approximately doubled its volume. The dough was then formed in a ball and placed on a baking parchment. After another hour of rest, it was baked for 25–30 min in a preheated oven at 200 °C. Between the end of the baking process and the consumption of the bread, there was a resting period of about 3 h.

#### 2.2.2. Test Product

The safety of alpha-cyclodextrin as a food additive and later as a food ingredient was assessed by the Joint Food and Agriculture Organization/World Health Organization FAO/WHO Expert Committee on Food Additives (JECFA) in June 2001 and June 2004. An Acceptable Daily Intake (ADI) “not specified” was allocated for both uses [7,8].

For the present study, food-grade alpha-cyclodextrin (Batch No 6P172) was obtained from Wacker Chemie, Munich, Germany. According to this manufacturer’s analytical data sheet, this batch had an alpha-cyclodextrin content of 99.1% on dry basis. The water content was 8.3% on as is basis (Karl Fischer method).

### 2.3. Study Design

The study was designed as single-blind study, i.e., the participants were unaware of the composition of the two test meals (aqueous solutions of alpha-cyclodextrin are tasteless) and the fact that a sequential treatment plan was applied. The protocol for measuring and calculating the blood glucose response was in line with the procedures recommended by the Food and Agriculture Organization/World Health Organization [6].

On three separate days, each participant received in the morning as breakfast, sequentially, (A) 50 g starch in the form of 100 g fresh white bread prepared daily as described in Section 2.2.1, together with 250 mL plain drinking water from the local water supply; (B) 50 g starch (100 g fresh white bread) together with 10 g alpha-cyclodextrin dissolved in 250 mL drinking water; and (C) 25 g alpha-cyclodextrin dissolved in 250 mL drinking water (no bread). The volunteers were instructed to consume the test meals at a comfortable pace within 15 min. Between treatment days there was a “wash-out” period of at least two days. During the 3-h period following the intake of the test meals, the participants remained seated at a table. The consumption of additional water was allowed, but it was recommended not to drink more than 300 mL.

### 2.4. Blood Sampling and Analytical Methods

After overnight fasting for at least ten hours, the subjects arrived between 8.00 and 8.30 a.m. at the clinic. A catheter was placed in the forearm vein for collection of venous blood. Capillary blood was collected from the fingertip in fluorocitrate tubes. Both venous and capillary blood samples were collected just before the consumption of the test meal (*t* = 0) and at 15, 30, 45, 60, 75, 90, 120, 150, and 180 min thereafter. Capillary whole blood was used for the glucose determinations, because the rise in blood glucose is said to be greater in capillary blood than in venous blood and because the results are less variable [4,5]. Glucose was measured in capillary blood using an automated hexokinase method (Olympus No. OSR 6121).

Plasma was prepared from the venous blood samples and stored at ˂−18 °C for about three weeks until insulin was determined in all samples at the same time using an automated radioimmunoassay (DRG Diagnostics Cat. No. EIA 2935). These analyses were performed by AlterChem, Agios Lavras Street 75a, GR-13231 Petroupoli, Greece. The results are expressed in μIU/mL (4 ng/mL = 100 μIU/mL).

### 2.5. Statistical Analyses

The incremental area under the curve (iAUC) of each subject’s blood glucose and insulin response was calculated according to the trapezoidal rule with baseline value (*t* = 0) as lowest level [6]. Results are given, as appropriate, as mean values ± standard deviation (SD) or mean values ± standard error of mean (SEM). Comparison of the treatment groups A and B was performed by repeated-measures analysis of variance (ANOVA) and paired t-test with a significance level of *p* < 0.05 for treatment B vs. treatment A. The spline function of R was used to calculate the interpolation of glucose and insulin values. Statistical analyses were performed using R version 3.6.0: a language and environment for statistical computing. (R Core Team, 2019), R Foundation for Statistical Computing, Vienna, Austria, without additional library packages.

## 3. Results

### 3.1. Participants and Compliance

The twelve volunteers who entered the study completed it according to protocol. Compliance with the requirement of overnight fasting prior to test days was good, as evidenced by the low basal blood glucose and insulin levels at the start of each treatment (Table 1). There were no significant within-person differences of baseline values.

### 3.2. Postprandial Glucose and Insulin Response

The ingestion of about 50 g starch in the form of 100 g fresh white bread (treatment A) resulted in the expected rapid rise in capillary blood glucose concentrations reaching the highest level of 8.73 ± 0.60 mmol/L after 53.8 ± 7.7 min. At the end of the postprandial 3-h observation period, the blood glucose concentrations returned to baseline values. Drops below starting levels (rebound effect) were small and were observed in 8 out of 12 subjects between 120 and 180 min after intake of the test meal.

When the fresh white bread was consumed together with 10 g alpha-cyclodextrin (treatment B), a markedly reduced and delayed glycemic response was observed, with a calculated maximum capillary blood glucose concentration of 6.19 ± 0.36 mmol/mL reached at 73.8 ± 19.7 min after consumption of the test meal.

The consumption of 25 g alpha-cyclodextrin with the drinking water (treatment C) was associated with an only small increase of blood glucose levels at 60 min after the test meal. At all other time points, blood glucose levels were not different from the baseline value, i.e., alpha-cyclodextrin did not induce a relevant glycemic response.

The time course of the calculated mean blood glucose values for the three treatments is depicted in Figure 1. The time courses of each individual’s blood glucose on each treatment are shown in the Appendix A.

The calculated mean plasma insulin levels rose after consumption of white bread (Treatment A) in the expected way (Figure 2). The addition of alpha-cyclodextrin to white bread (Treatment B) lead to a significantly lower insulin release, which, moreover, occurred with some delay. The intake of 25 g alpha-cyclodextrin on its own (Treatment C) had no effect on plasma insulin levels. Maximum blood insulin levels were reached with a median of about 45 min after consumption of white bread (treatment A), but with a median of about 75 min after consumption of white bread with alpha-cyclodextrin (treatment B).

Both the glycemic and insulinemic index of the applied white bread consumed together with 10 g alpha-cyclodextrin in the drinking water was, on average, about 43% relative to that of the white bread consumed with plain drinking water (Figure 3). The time courses of each participant’s blood insulin on each treatment are shown in the Appendix A.

### 3.3. Side-Effects

Adverse side-effects were not reported during the periods of bread consumption with or without alpha-cyclodextrin. However, three subjects reported mild bloating and one subject mild diarrhea after completion of treatment C. None of the subjects complained, however, about flatulence or intestinal cramps.

## 4. Discussion

The results of this study demonstrate that the consumption of 10 g of alpha-cyclodextrin together with fresh white bread, providing 50 g starch, reduces the post-prandial glycemic and insulinemic response significantly in healthy subjects. This is a beneficial health effect because post-prandial glycemia is related to body fat deposition and the risk of developing certain chronic diseases [9,10] while, vice versa, a low-glycemic diet is favorable for vascular health [11].

In several in vitro studies it has been shown that alpha-cyclodextrin inhibits the hydrolysis of amylose by porcine pancreatic alpha-amylase, presumably due to the competitive binding of alpha-cyclodextrin to the enzyme’s active site [12,13,14,15,16]. The results of the present study can be explained by this mechanism. However, it may be modulated by other components of more complex foods, such as fat that retards gastric emptying and that, by inhibiting the release of insulin after a meal, may blunt the blood glucose lowering effect of alpha-cyclodextrin [17,18,19].

Additionally, consistent with the concept of a specific amylase-inhibitory effect of alpha-cyclodextrin, rather than an unspecific effect (e.g., on gastric emptying time) is the observation that alpha-cyclodextrin, given at a dose of 10 g, did not reduce the glycemic response to sucrose (100 g single dose ingested with water) in healthy adults [20].

In comparison with other dietary fibers that have been examined for their glycemia lowering effect when added to bread, alpha-cyclodextrin appears to be particularly efficient [21]. Moreover, and in contrast to some other dietary fibers, the incorporation of alpha-CD in foods and beverages does not pose any particular food-technological or gustatory challenges (e.g., taste, viscosity).

Taking the published results into account, the European Food Safety Authority (EFSA) already earlier expressed a favorable opinion on the blood-glucose-lowering health effect of alpha-CD when consumed with a starch-containing meal [22].

## 5. Conclusions

This study demonstrates that the incorporation of alpha-cyclodextrin in a starch-based meal can significantly reduce its postprandial glycemic and insulinemic impact. Further studies are needed to evaluate the health benefits that may result from the regular consumption of foods containing alpha-cyclodextrin over longer periods of time, especially in populations at risk for developing diabetes and cardiovascular disease.

## Figures and Tables

**Figure 1 foods-09-00062-f001:**
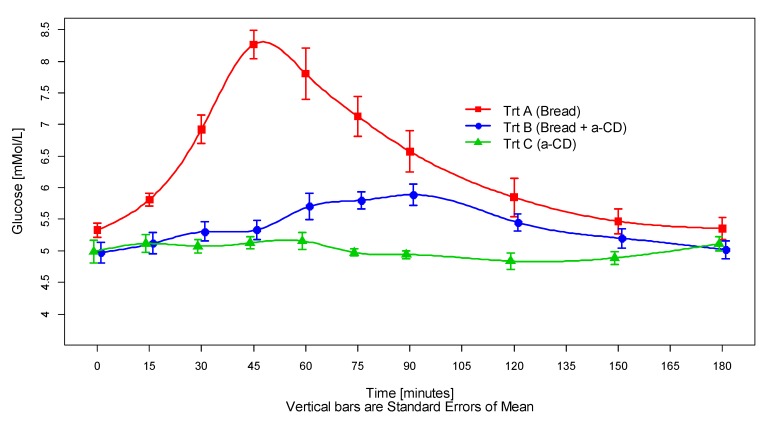
Glycemic response curves elicited by 100 g white bread consumed with 250 mL plain drinking water (treatment A), or 100 g white bread consumed with 250 mL water containing 10 g dissolved alpha-cyclodextrin (treatment B) or 250 mL water with 25 g dissolved alpha-cyclodextrin (alpha-CD) only (treatment C). Values are the means of the 12 subjects with their SEMs represented by vertical bars. Abbreviations: Trt, treatment; a-CD, alpha-cyclodextrin.

**Figure 2 foods-09-00062-f002:**
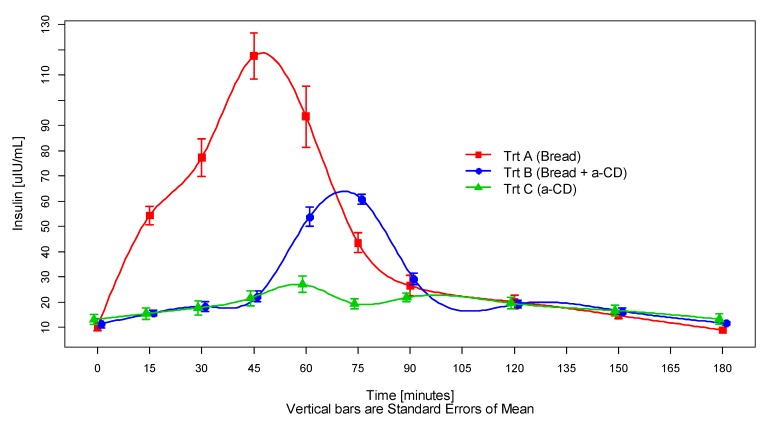
Insulinemic response curves elicited by 100 g white bread consumed with either 250 mL plain drinking water (treatment A), or 250 mL drinking water containing 10 g dissolved alpha-cyclodextrin (treatment B), or 250 mL drinking water containing 25 g dissolved alpha-CD only (treatment C). Values are the means of 12 subjects with their SEMs represented by vertical bars.

**Figure 3 foods-09-00062-f003:**
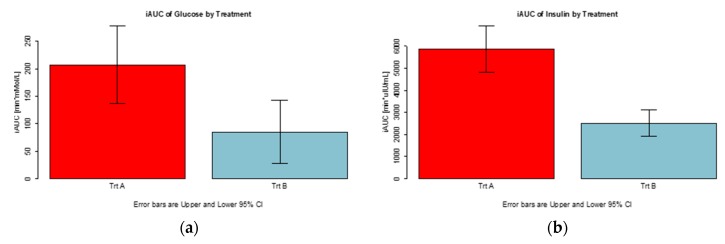
Integrated areas under curve (iAUC) of the blood glucose and insulin concentrations elicited over a period of 180 minutes by 100 g white bread consumed with either 250 mL plain drinking water (treatment A) or 100 g white bread consumed with 250 mL drinking water containing 10 g dissolved alpha-cyclodextrin (treatment B). Values are the means for 12 subjects with their SEMs represented by vertical bars. (**a**) integretated area under curve of blood glucose by treatment, (**b**) integrated area under curve of blood insulin by treatment.

**Table 1 foods-09-00062-t001:** Calculated mean blood glucose and insulin and insulin concentrations of all subjects at the start. (*t* = 0 min) of each treatment period.

Treatment	N	Glucose (mmol/L)			Insulin (μIU/mL)		
		Mean	95% CI		Mean	95% CI	
			Lower	Upper		Lower	Upper
A	12	5.33	5.09	5.57	10.3	6.5	13.0
B	12	4.97	4.62	5.33	11.6	7.6	15.6
C	12	4.99	4.60	5.38	13.1	8.6	17.7

Treatment A: 50 g starch (as 100 g fresh white bread) together with 250 mL drinking water. Treatment B: 50 g starch (as 100 g fresh white bread) together with 10 g alpha-cyclodextrin dissolved in 250 mL drinking water. Treatment C: 25 g alpha-cyclodextrin dissolved in 250 mL drinking water. Abbreviations: CI, Confidence Interval.

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
