# Peer review of "Alpha-Cyclodextrin Attenuates the Glycemic and Insulinemic Impact of White Bread in Healthy Male Volunteers"

_foods, 2020, doi:10.3390/foods9010062_

Round 1

Reviewer 1 Report

This study studies the effect of alpha-cyclodextrin on the postprandial glycemic and insulinemic effect of starch. Twelve male participants were included and the authors concluded that alpha-cyclodextrin in a starch-based meal can significantly reduce its postprandial glycemic and insulinemic impact.

This can potentially increase health benefits of starchy foods where this compound is added to.

The manuscript is well written, but several issues need to be addressed .

Major

Page 2, Recruitment: Male volunteers are used for the current study. Sex differences in glucose homeostasis exist however. This can have an effect on the generalisability of the results. Can the authors comment on this, incl future research in female participants and why this might be important? This also needs to be reflected in the conclusion of the article.

Page 2, Study design: The authors use a single-blinded trial with a sequential treatment plan. This has the advantage that each person is it’s own control and reduces confounding of age and sex. This is a little similar to a cross-over trial, but not completely. The authors only test one order of treatments: A, B and then C. However, treatment and period effects might play a role, hence, a proper cross-over trial - where different orders of treatment are tested - would be an appropriate next step. So, the current study seems to be more like a pilot and I suggest the article should reflect that.

Page 3, statistical analyses:

(1) is a power calculation conducted? If this study is a pilot study, this is not necessary. But if it’s not, than a power calculation is good common practice and might explain - not in this case - why no significant difference is observed.

(2) was the repeated measures Anova used to test A, B and C simultaneously? And the paired t-test only for A vs B? If yes, this needs to be stated as it’s not completely clear what is used for what. Also, for reparations measures Anova, where post-hoc tests ( basically paired t-tests with nonferrous correction applied to p-values) used when overall test showed significance?

(3) re the spline analysis, what type of spline was used, B-spline, cubic, other? Where number of knots and positions chosen? This information needs to be included.

(4) were the BL values for each treatment (A, B or C) tested whether these were different within the persons?

Page 3-4, Results

More information is needed on effect measures, p-values and when results come from the repeated measures Anova or from a post hoc test etc. Also, how are the spline analysis applied for this study. The effect measures, p-values also need to be added to the abstract.

Minor

Page 2, test product: does the water with or without the test product taste differentially? And how has this been established?

Page 2, line 76: the authors wrote one as is basis’. This does not seem correct.

Page 2, study design: how have the authors established that the wash-out period of 2 days is long enough? Could the authors elaborate on this?

Page 3, analytical methods: Could the authors provide some more information on the methods used (eg, calibration, variation)?

Page 4, lines 136-138: only here the readers are referred to Figure 1, suggest to do already at bottom of page 3, when the authors are starting to describe the results from this figure.

Appendix A: the size of the markers in the lines are quite small, which makes it hard to differentiate between them. Could the size of the markers be increased?

Author Response

Page 2, recruitment:

This study is considered to be a pilot study and for this reason, only male volunteers were included. This is now properly indicated in the manuscript. Moreover, it should be taken into account that alpha-cyclodextrin acts by competitively inhibiting alpha-amylase. This effect is highly unlikely to be modulated by the sex of the volunteer. 

Page 2, study design:

We agree that in the future, a more definitive study should be designed as a crossover study. As suggested, we have now properly described the here reported study as a pilot study.

Page 3, statistical analysis:

(1) This study is now clearly described as a "pilot study".

(2) The statistical comparison pertains to group A and group B only. Group C was included only for examining any (unlikely) direct effect of alpha-CD (ingested as such) on the glycemic response. 

(3) For the graphical presentation the means at each time point (Figures 1 and 2), the points were connected through interpolation by a cubic spline function. 

(4) The baseline values were tested. There were no significant within-person differences.

Page 3-4, results:

Confidence intervals were reported. The AUC values were calculated by the trapezoidal method of each volunteer and then the mean and standard error was calculated for each treatment group. 

Minor

Page 2:

Alpha-cyclodextrin as such does not affect the taste of the water. 

Page 2, line 76:

The term "on as is basis" refers to the crystalline alpha-cyclodextrin as it has been obtained from the supplier and was used in the reported study. The reported dose (and thus intake) of alpha-cyclodextrin pertains to the product as received, i.e. the dose on a dry matter basis would be 9.17 g (rather than 10 g).

Page 2, study design: 

Since alpha-CD acts on pancreatic amylase in the lumen of the small intestine and since the volunteers were following normal eating habits between the treatments, it is highly unlikely that there be any carry-over of alpha-cyclodextrin from the first to the second treatment period. In addition, group A was bread only without CD anyway!

Page 3, analytical methods:

Standard methods were applied by the laboratory that performs such tests routinely. The test results are therefore considered to be valid for the purpose of this study. 

Page 4, line 136-138:

The reviewer's suggestion has been followed. 

Appendix A: 

We increased the size of the markers. 

Reviewer 2 Report

This study examining the effect of alpha-cyclodextrin on postprandial glycemic and insulinic load of a food equivalent of 50 g of starch in 12 healthy male volunteers.
The submitted manuscript is considered of interest to the readers of the magazine, despite presenting a small sample composed only of men, an aspect that should be argued in the discussion and nuanced in the conclusions. The results should not be generalized to both sexes since it is not derived from this research study. Authors should always clarify that the results refer to adult males, which is what has been studied. The authors must argue the possible alteration of the results due to the small sample size. Chart: In figure 1 and figure 2 must be corrected “vertikal bars” by “vertical”.

Author Response

We have followed the reviewer's advice and have not generalized the results to males and females. However, the reviewer would probably agree that the inhibitory effect of alpha-CD on alpha-amylase is not influenced by the sex of the volunteer. 

This study is considered to be a pilot study and for this reason, only male volunteers were included. This is now properly indicated in the manuscript. Moreover, it should be taken into account that alpha-cyclodextrin acts by competitively inhibiting alpha-amylase. This effect is highly unlikely to be modulated by the sex of the volunteer. 

Page 2, study design:

We agree that in the future, a more definitive study should be designed as a crossover study. As suggested, we have now properly described the here reported study as a pilot study.

Reviewer 3 Report

This manuscript addresses an important area of research.

Comments:

1. Could the authors please clarify how the number of study participants were determined i.e. how was the study powered?

2. Parametric data analyses were performed; were the data normally distributed? The number of study participants are small. 

Author Response

(1) The study has been planned as a pilot study (as it is now indicated in the manuscript). In view of the results of earlier studies on the effects of dietary fibers on the glycemic response to high glycemic meals, twelve subjects were considered to suffice.

 (2) Yes, the data were normally distributed as can be seen in the plots of Appendix A.

Round 2

Reviewer 1 Report

No further comments

Reviewer 3 Report

The authors have provided a satisfactory response to my comments.